# Enhancing Variational Quantum Algorithms: Effective Quantum Ansatz Design Using GFlowNets

## Abstract

Quantum computing promises significant computational advantages over classical computing. However, current devices are constrained by a limited qubit count and noise. By combining classical optimization methods with parameterized quantum circuits, Variational Quantum Algorithms (VQAs) offer a potential solution for noisy intermediate-scale quantum systems (NISQ). This makes VQAs particularly promising strategies for achieving near-term quantum advantages; such approaches are now widely explored for nearly all quantum computing applications. However, designing effective parameterized circuits, also known as ansatz, remains challenging. In this work, we introduce the use of GFlowNets as an efficient method to automate the development of efficient ansatz for various quantum computing problems. Our approach leverages GFlowNets to efficiently explore the combinatorial space of parameterized quantum circuits. Our extensice experiments demonstrate that GFlowNets can discover ansatz with an order of magnitude fewer parameters, gate counts, and depths compared to current approaches for the molecular electronic ground state energy problem. We also apply our approach to the unweighted Max-Cut problem, where we observe similar improvements in circuit efficiency. These results highlight the potential of GFlowNets to significantly reduce the resource requirements of VQAs while maintaining or improving solution quality.

## 1 Introduction

Quantum computing has made remarkable progress in the past decade. Novel quantum devices boast high qubit count and ever improving coherence times (Preskill, 2018; Farhi et al., 2014; Peruzzo et al., 2014; Lau et al., 2022; Li et al., 2020; Chow et al., 2021). Despite these promising advances, we remain in the era of noisy intermediate-scale quantum (NISQ) systems: modern quantum computers are not yet fully fault tolerant and are still sensitive to noise. Many research efforts have been made in order to render NISQ-era quantum devices usable for practical purposes (Bang & Yoo, 2014; Farhi et al., 2014; Lamata et al., 2018; Lau et al., 2022; Peruzzo et al., 2014; Li et al., 2020). Variational quantum algorithms (VQAs), for instance, show promise in this direction. VQAs are hybrid algorithms which use classical expectation-maximization (EM) methods in tandem with quantum circuits to iteratively fit parameterized circuits. These methods can be used to solve a variety of problems, such as the ground state energy problem, quantum machine learning, and combinatorial optimization problems, which have applications in many fields, e.g. molecule discovery, logistics, and machine learning (Peruzzo et al., 2014; Farhi et al., 2014; Li et al., 2020; Biamonte et al., 2017; Schuld & Killoran, 2019; Schuld, 2021; Dai & Krems, 2022; Guo et al., 2024; Torabian & Krems, 2023; Cerezo et al., 2022).

Despite their promise, one of the main challenges in VQA research lies in designing efficient quantum circuits (or ansatz); the space of possible circuits is inherently combinatorial and it is impossible to explore it fully (Peruzzo et al., 2014; Farhi et al., 2014). Furthermore, many current approaches to ansatz design either rely on human expertise or use heuristic-based methods, which may not scale well as problem sizes grow.

In this work, we propose to use GFlowNets (GFNs) (Bengio et al., 2023) to automate quantum circuit design. GFlowNets are powerful generative models based on stochastic reinforcement learning (RL).

GFNs are particularly well-suited for this task because they can learn a policy to sample complex, compositional structures, such as quantum circuits, while promoting diversity in the generated solutions (Bengio et al., 2021). By sampling candidate solutions *proportionally to the reward distribution*, they can uncover the underlying structure of a solution space, leading to efficient exploration even when the number of solutions is prohibitively high.

Our approach fundamentally relies on treating circuit design as a *decision problem*. We leverage the compositional structure of quantum circuits and the diversity-seeking policy of GFNs to generate diverse and efficient circuit structures for the problem at hand. This enables the discovery of ansatz with significantly fewer parameters, gate counts, and depths compared to current approaches, as demonstrated by our experimental results. Notably, for the molecular electronic ground state problem, our approach achieve circuits that require an order of magnitude fewer resources without sacrificing accuracy. Similarly, for the unweighted Max-Cut problem, we observe substantial improvements in circuit efficiency.

**Key Contributions**

- We develop and apply GFlowNets for efficient quantum circuit sampling in quantum chemistry and combinatorial optimization problems.

- Our method achieves energy estimates for molecular ground state problems within chemical accuracy and successfully solves the unweighted Max-Cut problem, while significantly reducing circuit complexity compared to alternative approaches.

- We demonstrate the generalizability of GFlowNet-sampled circuits, which maintain accuracy when re-optimized for different molecular geometries not encountered during training.

The rest of this work is organized as follows. In Section 2, we discuss related works on quantum circuit design and GFlowNets. In Section 3, we introduce GflowNets and variational quantum algorithms for ground state energy problem and unweighted max cut problem. In Section 4, we present the methodology for the quantum architecture search process, including the definitions of states, actions, and rewards. In Section 5, we describe the experimental setup, detailing how the models were evaluated on the ground state energy problem and the unweighted Max-Cut problem. In Section 6, we present and discuss the results, analyzing the performance of our approach on both problems, followed by conclusions and future work in Section 7.

## 2 RELATED WORK

**Quantum Circuit Design** Evolutionary algorithms, particularly genetic algorithms (GA) have been applied to quantum circuit design since the late 1990s, with notable success in evolving simple circuits like quantum error correction codes (Williams & Gray, 1998; Massey et al., 2004; Bang & Yoo, 2014; Lamata et al., 2018; Las Heras et al., 2016). However, GAs are limited by gene length, candidate gate set size, and their inability to handle parameterized rotation gates, reducing their effectiveness for evolving VQA ansatz. Reinforcement learning algorithms, such as those by (Ostaszewski et al., 2021; Ye & Chen, 2021; Kuo et al., 2021) have shown promise in selecting gates for VQE, outperforming chemistry-inspired methods like UCCSD (Romero et al., 2018) in efficiency and accuracy. RL approaches have demonstrated success on smaller systems. Sampling-based algorithms introduce adaptive VQE ansatz searches (Grimsley et al., 2019; Tang et al., 2021; Du et al., 2022; Zhang et al., 2022; Wu et al., 2023; Lu et al., 2023), though they suffer from inefficiencies and require predefined circuit structures, limiting their flexibility. The Generative Quantum Eigensolver (GQE) (Nakaji et al., 2024), and specifically its transformer-based variant GPT-QE, leverages generative models and pre-training for quantum circuit generation. While effective in finding ground states for electronic structure Hamiltonians, GQE is constrained by using predetermined quantum circuit structures as tokens, leading to long circuits that can reduce efficiency and flexibility in broader applications. Another work (Fürrutter et al., 2024) use text-conditioned denoising diffusion models to efficiently generate quantum circuits, overcoming simulation bottlenecks and advancing entanglement generation and unitary compilation tasks. Other approaches such as Brute Force Search (Torabian & Krems, 2023; Guo et al., 2024) and Bayesian Optimization (Duong et al., 2022) are shown to lead to less efficient search processes, requiring large computational resources.

**GFlowNets** GFlowNets have have been used successfully across a number of disciplines such as drug discovery (Bengio et al., 2021), phyllogenetic inference (Zhou et al., 2023), causal graph learning (Nishikawa-Toomey et al., 2022) and even language model finetuning (Hu et al., 2023). GFlowNets are based in a rich literature ranging from sampling methods to stochastic RL. Prior to GFlowNets, many MCMC methods (Grathwohl et al., 2021; Dai et al., 2020) were designed to sample with probability proportional to some unnormalized energy function. However, the problem of mixing between modes remains challenging for such methods. (Buesing et al., 2020) propose a method to *train a policy* to generate samples proportional to some reward. However, their method only works in the setting where a single path leads to a given state. Moreover, many connexions have been drawn between GFlowNets and other well-known methods. In terms of RL, the GFN objective may be reminescent of soft Q-Learning (Haarnoja et al., 2017). In fact, Tiapkin et al. (2024) show that GFlowNets are formally equivalent to stochastic RL with entropy regularization. It is also possible to treat the GFlowNet learning problem as a convex MDP problem (Zahavy et al., 2021) with the goal of minizing KL divergence. Finally, Malkin et al. (2022b) show that variational inference algorithms are equivalent to special cases of GFlowNets.

## 3 PRELIMINARIES

In this section, we give a cursory introduction to GFlowNets and Variational Quantum Algorithms. We also introduce the Unweighted Max-Cut problem and show how to utilize quantum computing to solve it.

### 3.1 GFLOWNETS

GFlowNets (Bengio et al., 2021; 2023) are a family of generative models which sample compositional objects $x$ such as graphs or strings. To generate such objects, GFlowNets learn a stochastic policy $\pi(x)$ that generates samples proportionally to some reward function $R(x)$. Every step of this stochastic decision process can be seen as adding a component to $x$. For example adding an edge or a node to a graph.

**Problem Setting** Let $G = (\mathcal{S}, \mathcal{A})$ be a directed acyclic graph (DAG) with state space $\mathcal{S}$ and action space $\mathcal{A}$. we will denote by $s_0 \in \mathcal{S}$ the *initial* state (a state with only outgoing transitions) and we will denote by either $s_f$ or $x \in \mathcal{X} \subset \mathcal{S}$ a terminating state (which only has incoming transitions).

The goal of a GFlowNet is to iteratively construct an object $x \in \mathcal{X}$ by starting from an initial state $s_0$ and sequentially applying actions until it reaches a terminating state $s_f$ (or $x$). The sequences of steps taken to construct $x$ are defined as a trajectory $\tau = (s_0 \to s_1 \to ... \to s_f)$. This sequence can be thought of as a path on the DAG. We denote by $\mathcal{T}$ the set of all such trajectories from $s_0$ to some terminating state.

The value of the constructed object $x$ is measured through some reward function $R : \mathcal{X} \to \mathbb{R}^+$. This reward function usually takes the form $R(x) = e^{-\beta \mathcal{E}(x)}$, where $\mathcal{E}$ is some energy function and $\beta$ is some temperature parameter.

**Flows and flow matching** The main intuition behind the GFlowNet formulation is to consider the DAG as a flow network, where the flow through some node (or edge) corresponds to some unnormalized probability of visiting said node (or edge respectively). This notion can be formalized through the definition of *flow functions* (Bengio et al., 2021). We start by defining the *trajectory* flow $F : \mathcal{T} \to \mathbb{R}^+$, where $F(\tau)$ represents the unnormalized probability mass corresponding to trajectory $\tau$. The *state* flow is thus defined by $F(s) = \sum_{s \in \tau} F(\tau)$ (i.e. the sum of all trajectory flows passing through $s$). Equivalently, we define the *edge* flow $F(s \to s') = \sum_{s \to s' \in \tau} F(\tau)$ (i.e. the sum of all trajectory flows which pass through edge $s \to s'$).

Flow functions induce a measure over $\Omega = 2^{\mathcal{T}}$. By introducing some normalizing constant $Z$, we can induce a probability measure $P(\tau) = F(\tau)/Z$. This lets us define the forward transition probability

$$P_F(s \mid s') = \frac{F(s \to s')}{F(s)}, \tag{1}$$

which is crucial to the implementation of GFlowNets. Finally, we say a flow is *consistent* if $\forall s \in \mathcal{S}$, we have:

$$\sum_{s' \in \text{Parent}(s)} F(s' \rightarrow s) = \sum_{s'' \in \text{Child}(s)} F(s \rightarrow s'') \tag{2}$$

As it is shown in (Bengio et al., 2021), any consistent flow with terminal flow set to $R(x)$ induces a forward policy $P_F(s' \mid s)$ which samples objects proportionally to $R(x)$.

**Training GFlowNets**   The main idea behind training GFlowNets is to learn a policy which has a consistent flow. This policy is usually learnt by some form of neural network which has enough capacity for the task at hand. We constrain the policy to learn a consistent flow by enforcing flow matching in the loss function. The original flow matching loss given by (Bengio et al., 2021) is defined as

$$\mathcal{L}_{FM}(s; \theta) = \left( \log \frac{\sum_{s' \in \text{Parent}(s)} F(s' \rightarrow s)}{\sum_{s'' \in \text{Child}(s)} F(s \rightarrow s'')} \right)^2 \tag{3}$$

This can be though of as minimizing the log MSE of the incoming and outgoing flows presented in Eq. (2).

**Trajectory Balance**   In this paper, we consider *trajectory balance* (Malkin et al., 2022a) as our main objective function. This objective is defined as

$$\mathcal{L}_{TB}(\tau; \theta) = \left( \log \frac{Z_\theta \prod_{s \rightarrow s' \in \tau} P_F(s' \mid s; \theta)}{R(x) \prod_{s' \rightarrow s \in \tau} P_B(s \mid s'; \theta)} \right)^2 \tag{4}$$

Here, both $P_F$ and $Z_\theta$ are learnt. This method has shown faster credit assignment when learning GFlowNets and is robust to long trajectories, which motivates its use throughout this work.

## 3.2 Variational Quantum Algorithms (VQAs)

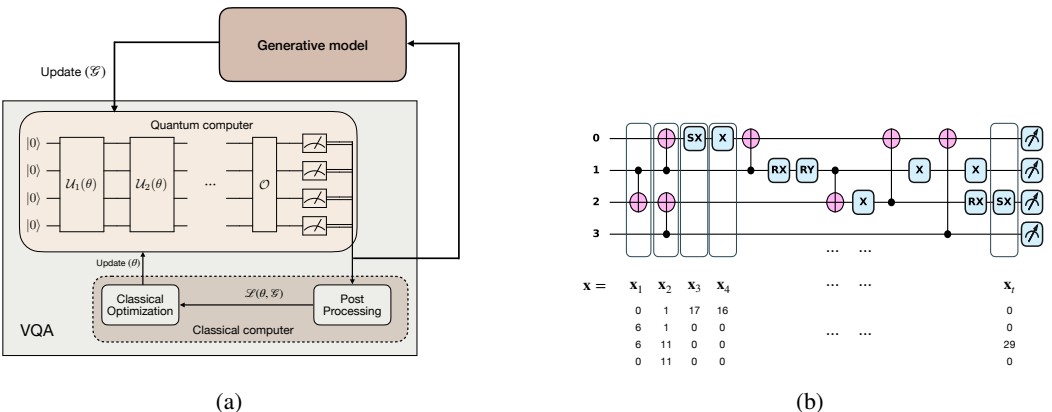

(a)                                      (b)

Figure 1: (a) Schematic representation of a Variational Quantum Algorithm (VQA) in the context of Quantum Architecture Search (QAS). The quantum computer is initialized with qubits in the $|0\rangle$ state, followed by the application of parameterized quantum gates $\mathcal{U}_1(\theta), \mathcal{U}_2(\theta)$ ... , and an observable measurement $\mathcal{O}$. The measurement results are fed into a classical computer for post-processing, including classical optimization and generative model updates. The optimized parameters $\theta$ and generative model are iteratively refined to minimize the task-dependent loss function. GFlowNets are utilized to search for optimal quantum operation compositions. (b) Example of input data representation for a sequence model in QAS. The sequence $\mathbf{x}$ consists of elements $\mathbf{x}_1, \mathbf{x}_2, \mathbf{x}_3, \ldots, \mathbf{x}_t$. Each $\mathbf{x}_i$ is a vector with a length corresponding to the number of qubits, representing which gates are applied to each qubit at a given timestep in the sequence. Gates are represented using tokenization: each operation has an associated integer value. 0 represents the identity gate.

VQA are hybrid quantum-classical methods that optimize parameterized quantum circuits to minimize an objective function. As shown in Fig. 1a, the first step of VQA is to define a loss function $\mathcal{L}$,

which encodes the objective of the algorithm, and a parameterized quantum circuit (or ansatz) with parameters $\theta$ (Peruzzo et al., 2014; Farhi et al., 2014; Cerezo et al., 2021). The main objective of this approach is to solve the optimization problem $\theta^* = \arg\min_\theta \mathcal{L}(\theta)$. Generally, the loss function can be expressed as

$$\mathcal{L}(\theta) = \sum_i f_i(\mathrm{Tr}(\mathcal{O}_i U(\theta)\rho_i U^\dagger(\theta))), \tag{5}$$

where $\rho_i$ are input states, $\mathcal{O}_i$ are observations with $\mathcal{O}_i$ hermitian (*i.e.* $\mathcal{O}_i^\dagger = \mathcal{O}_i$), $U(\theta)$ is a parameterized unitary with parameters $\theta$ and $f_k$ is a set of functions relative to the task at hand. We use $\mathrm{Tr}$ to denote the trace operator and $U^\dagger$ to denote the conjugate transpose of a some unitary $U$.

The Quantum Architecture Search (QAS) problem for VQAs can be formally expressed as follows. Given a set of quantum gates $\mathbf{G} = \{g_1, g_2, \ldots\}$, where each $g_i$ represents a candidate quantum gate acting on an arbitrary number of qubits, the goal of QAS is to explore all possible compositions of these gates. Each composition $\mathcal{G}$ is a sequence of gates $g_{i_1} \circ g_{i_2} \circ \cdots$, which forms a candidate quantum circuit. The objective is to find the optimal composition $\mathcal{G}^*$, along with the corresponding parameterized unitary $U(\mathcal{G}^*, \theta^*)$, that minimizes the loss function $\mathcal{L}$ of some given VQA task. Formally this can be expressed as,

$$\mathcal{G}^*, \theta^* = \arg\min_{\mathcal{G}, \theta} \mathcal{L}(\mathcal{G}, \theta). \tag{6}$$

In this work, we apply the QAS framework to two well-known tasks: the ground state energy problem and the unweighted Max-Cut problem. Below, we provide an overview of each problem and describe how they fit within the VQA and QAS methodology.

**Ground state energy problem** One of the key applications of the VQA framework is the Variational Quantum Eigensolver (VQE) (Peruzzo et al., 2014), which is designed to find the ground state energy, or lowest energy, $E_0$, of a given Hamiltonian $\mathcal{H}$. This method has its roots in the variational principle of quantum mechanics, which guarantees that the expectation value of the Hamiltonian $\mathcal{H}$, calculated with any trial quantum state $\rho(\theta) = U(\mathcal{G}, \theta)\rho_0 U^\dagger(\mathcal{G}, \theta)$, will always be lower bounded by the ground state energy $E_0$.

The optimization task can be framed similarly to the general QAS problem. The goal is to find the optimal circuit architecture $\mathcal{G}$ and parameters $\theta$ that minimize the expectation value of the Hamiltonian,

$$E_{\mathrm{approx}} = \min_{\mathcal{G}, \theta} \mathrm{Tr}(\mathcal{H}U(\mathcal{G}, \theta)\rho_0 U^\dagger(\mathcal{G}, \theta)) \geq E_0. \tag{7}$$

This formulation of the ground state energy problem fits naturally into the VQA and QAS framework, where the task is to efficiently explore the space of quantum circuits to find an optimal solution.

**Unweighted Max-Cut** Another key application of the Quantum Architecture Search framework is solving classical combinatorial optimization problems, such as the unweighted Max-Cut problem, which is known to be NP-hard (Karp, 2010). In this problem, we consider an unweighted graph $\mathcal{G} = (\mathcal{V}, \mathcal{E})$ where $\mathcal{V}$ is a set of vertices, and $\mathcal{E}$ a set of edges. The objective is to partition the vertex set $\mathcal{V}$ into two disjoint subsets $\mathcal{V}_0$ and $\mathcal{V}_1$ such that $\mathcal{V}_0 \cup \mathcal{V}_1 = \mathcal{V}$ and $\mathcal{V}_0 \cap \mathcal{V}_1 = \emptyset$, maximizing the number of edges between $\mathcal{V}_0$ and $\mathcal{V}_1$. With a quadratic programming approach, the problem can be formulated as follows,

$$\max \sum_{0 < i < j \leq n} w_{i,j} \frac{1 - x_i x_j}{2}$$
$$\text{s.t. } x_k \in \{-1, 1\}, \quad 1 \leq k \leq n \quad , \tag{8}$$
$$w_{i,j} = \mathbf{1}_{e_{i,j} \in \mathcal{E}}, \quad 1 \leq i < j \leq n$$

where $n$ is the number of vertices. To utilize quantum computing for solving the unweighted Max-Cut, we reformulate the problem using a matched observable $\mathcal{O}_c$, where each qubit corresponds to a vertex in the graph (Farhi et al., 2014). The observable can be calculated as

$$\mathcal{O}_c = \sum_{e_{i,j} \in \mathcal{E}} \frac{1}{2}(I - Z_i Z_j), \tag{9}$$

where $I$ is the identity matrix and $Z_i$ is the Pauli-Z matrix acting on qubit $i$. The eigenstate with the largest eigenvalue represents the solution to the Max-Cut problem.

Similar to the ground state energy problem in VQAs, the unweighted Max-Cut problem can also be framed as a QAS task, where the goal is to find the optimal quantum circuit $\mathcal{G}$ and parameters $\theta$ that minimize the corresponding loss function. This can be formulated as,

$$\mathcal{G}^*, \theta = \arg\min_{\mathcal{G}, \theta} -\mathrm{Tr}(\mathcal{O}_c U(\mathcal{G}, \theta)\rho_0 U^\dagger(\mathcal{G}, \theta)). \tag{10}$$

This illustrates how the QAS framework can be applied to classical optimization problems by leveraging quantum circuits to find optimal solutions efficiently.

## 4 METHODOLOGY

### 4.1 QUANTUM ARCHITECTURE SEARCH PROBLEM AS AN INFERENCE PROBLEM WITH GFLOWNETS

QAS problems for VQA involve optimizing a task-dependent loss function with discrete parameters $\mathcal{G}_i$ and continuous parameters $\theta$. In this work, we use GFlowNets to solve the discrete part of this optimization problem. The QAS problem involves discovering efficient circuit compositions for a given task, which requires exploring a combinatorial search space. We approach this by efficiently sampling from an energy-based model, where the probability distribution $p_T^*(\mathbf{x}) \propto \exp(-\beta\mathcal{E}(\mathbf{x}))$ guides the exploration of possible compositions $\mathcal{A}_i$. Here, $\beta$ is the inverse temperature parameter. Sampling a well-trained GFlowNet is equivalent to sampling from $p_T^*(\mathbf{x})$ itself. As $\beta$ approaches zero, the GFlowNet converges to the distribution of the optimal compositions of quantum operations. Conversely, as $\beta$ approaches infinity, the GFlowNet converges to the distribution of all possible compositions of quantum operations (Bengio et al., 2023). As stated in Section 3.1, learning to sample from this distribution is equivalent to learning a stochastic policy $\pi$ by flow-matching. Thus, we frame the QAS problem as an RL problem with the following constituents:

**State** The state of each VQA problem is defined by a composition, denoted as $\mathcal{G}$, which consists of quantum gates arranged into a quantum circuit. This circuit is characterized by a sequence $\mathbf{x} = \mathbf{x_1} \ldots \mathbf{x_t}$, where each $\mathbf{x_i}$ represents an action vector $\mathbf{x_i} = (x_i^1, \ldots, x_i^n)$. In this context, each element $x_i^j$ belongs to the set $0, 1, 2, \ldots, N_g$, specifying the quantum gate applied at time $i$ to qubit $j$, as shown in Figure 1b. For instance, $x_i^j = 0$ indicates that an identity gate is applied to qubit $j$. The symbol $N_g$ represents the total number of distinct quantum operations available. Each state represents a variational quantum circuit and is a terminal state. It is important to note that not all quantum operations are applied to every qubit at each time step; if no action is applied to a qubit, zero is recorded in the action vector.

**Action** We define a set of quantum gates $\mathbf{G} = \{X, SX, H, R_X, R_Y, R_Z, CNOT\}$. We assume that all single-qubit gates can be applied to any qubit. The CNOT gate requires one qubit to act as a control and another as a target; we include all possible pairings of qubits for this gate. With this choice of quantum gates, we have the necessary components for universal quantum computations as shown in Theorem 4.1. The subset $\{R_X, R_Y, R_Z\}$ consists of parameterized rotation gates, which introduce free parameters to the quantum circuits. The initial state is an empty sequence $\mathbf{x}_0 = \emptyset$, representing the quantum state beginning with the all-zero state $|0\rangle$. The action of the model is to choose a gate from $\mathbf{G}$ to act on the quantum state. Each selected gate from $\mathbf{G}$ modifies the quantum state. To prevent the model from selecting invalid or redundant actions, we apply a forward mask, as described in Appendix A.2.

**Theorem 4.1.** *(Nielsen & Chuang, 2001) The set of gates* $\mathbf{G} = \{X, SX, H, R_X, R_Y, R_Z, CNOT\}$ *is universal for quantum computation.*

**Reward** We set the log reward to be the loss of the VQA problem with scaling parameter $\beta$ and offset parameter $b$, *i.e.* $\mathcal{E}(x) = -\beta\min_\theta(\mathcal{L}(\mathcal{G}_i, \theta) - b)$. When a terminal state is reached, we optimize the parameters $\theta$ to minimize the loss function $\mathcal{L}$ associated with $\mathcal{G}$ in order to evaluate the reward.

## 5 EXPERIMENTS

### 5.1 EXPERIMENTAL SETUP

For the GFlowNet implementation, we use a transformer network with a hidden size of 4096, 4 layers, an embedding dimension of 256, and 8 attention heads for the forward policy. The backward policy is modeled by a uniform distribution to quickly approach one of the optimal solutions as suggested in (Malkin et al., 2022a). We implement all our models in PyTorch (Paszke et al., 2019), and we adopt trajectory balance as the main objective function. The model is updated every 5 iterations, after computing the batch loss for 5 complete trajectories. We use the Adam optimizer with a learning rate of $1 \times 10^{-4}$ for the forward policy model $P_F$ and $1 \times 10^{-1}$ for the partition function parameter $Z_\theta$. The inverse temperature parameter $\beta$ used in the reward function, is set to 1000 based on prior work by Zhang et al. (2023). We limit the circuits in all experiments to a maximum 20 parameters (number of rotation gates) and a maximum depth of 30. All quantum computations are performed using a noise-free simulator provided by PennyLane, a comprehensive software framework for quantum computing (Bergholm et al., 2018). We leverage PennyLane's state vector simulator to execute the variational quantum circuits. The circuit parameters $\theta$ are optimized using PennyLane's built-in Adam optimizer, with a learning rate of $1 \times 10^{-2}$, 10 random reinitializations, and a maximum of 2000 optimization steps.

### 5.2 GROUND STATE ENERGY PROBLEM

In this experiment, we aim to determine the ground state energies of the hydrogen ($H_2$) and lithium hydride (LiH) molecules. We perform calculations at various intramolecular bond distances using different approximation schemes, which result in Hamiltonians defined over different numbers of qubits. The computations are carried out using the PennyLane and PySCF plugins (Sun et al., 2018), and all calculations utilize the STO-3G basis set. For the hydrogen molecule ($H_2$), we compute the ground state energy using the full Hamiltonian, with no additional approximations. This results in a 4-qubit Hamiltonian. We set the offset parameter $b = 0$. We analyze the method's performance at two different H-H bond distances: 0.745Å and 1.5Å. For the lithium hydride (LiH) molecule, we apply an approximation that leverages the molecule's symmetry, and we remove a frozen orbital that interacts weakly with the other orbitals, as described in (Bravyi et al., 2017; Setia et al., 2020). This approximation reduces the complexity of the problem, resulting in a 6-qubit Hamiltonian. We set the offset parameter $b = -7$. We consider two Li-H bond distances: 1.4Å and 2.2Å. In all cases, we use the Jordan-Wigner transformation to map the fermionic Hamiltonians to qubit-based spin operators for quantum computation. This experimental setup allows us to evaluate the performance of the proposed method in determining the ground state energies for these molecular systems. As a reference, we also include results from Hartree-Fock (HF) (Szabo & Ostlund, 2012), a mean-field method providing an approximate solution, and UCCSD (Unitary Coupled Cluster with Single and Double excitations) (Romero et al., 2018), a standard chemistry-inspired ansatz, for comparison.

### 5.3 UNWEIGHTED MAX CUT PROBLEM

In the experiments conducted to solve the unweighted Max Cut problem, we evaluated performance, focusing on graphs generated by the Erdős–Rényi model with 10 vertices at varying edge creation probabilities ($p_e$). The mean values of key metrics such as the number of edges ($N_{edges}$), mean degree (D), minimum degree ($D_{min}$), and maximum degree ($D_{max}$) were reported, along with the Conditional Value-at-Risk (CVaR) metric and the depth of the solution quantum circuits. We set the offset parameters $b = 0$ for this problem.

## 6 RESULTS AND DISCUSSION

### 6.1 GROUND STATE ENERGY PROBLEM

Fig. 2 shows the estimated molecular energies obtained from VQE on the ground state energy problems. The circuits sampled by the GFlowNet model yield energy estimates that are within chemical accuracy when compared to the true ground state energy. The performance for $H_2$ and LiH indicates that the VQE energy estimated using GFlowNet-sampled circuits closely approximate the

Table 1: Energy error comparison in Hartree across different models, with chemical accuracy of $1.6 \times 10^{-3}$ Hartree.

| System | Method | Energy (Hartree) |
|--------|--------|------------------|
| $H_2$ | UCCSD (Romero et al., 2018) | $5.5 \times 10^{-11}$ |
| | Ours | $2.8(2.9) \times 10^{-9}$ |
| | QuantumDARTS (Wu et al., 2023) | $4.3 \times 10^{-6}$ |
| | QCAS (Du et al., 2022) | $2.2 \times 10^{-2}$ |
| | DQAS (Zhang et al., 2022) | $3.1 \times 10^{-4}$ |
| | CRLQAS (Patel et al., 2024) | $7.2 \times 10^{-8}$ |
| LiH | UCCSD (Romero et al., 2018) | $4.0 \times 10^{-5}$ |
| | Ours | $7.2(2.6) \times 10^{-4}$ |
| | QuantumDARTS (Wu et al., 2023) | $2.9 \times 10^{-4}$ |
| | QCAS (Du et al., 2022) | $7.3 \times 10^{-2}$ |
| | DQAS (Zhang et al., 2022) | $1.5 \times 10^{-3}$ |
| | CRLQAS (Patel et al., 2024) | $6.7 \times 10^{-4}$ |

exact energies across various bond lengths, as shown by the green and red markers. Our method achieves comparable results to QuantumDARTS (Wu et al., 2023),and CRLQAS (Patel et al., 2024) though with slightly larger error compared to UCCSD (Romero et al., 2018), as shown in Table 1. However, the energy estimates from our models remain well within chemical accuracy, which is promising for real chemical applications. A key distinction of our method is its ability to sample multiple circuits using the same model. Further details on training and sample accuracy are provided in Appendix D.

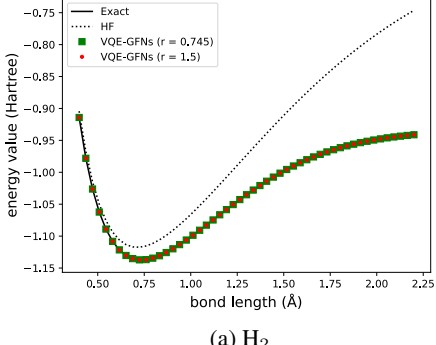
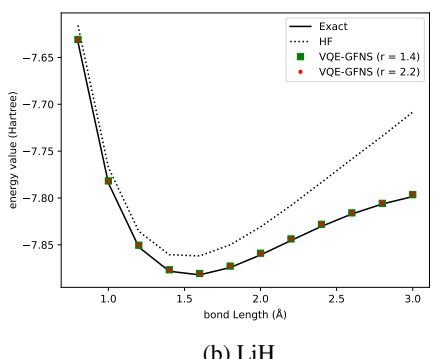

(a) $H_2$                    (b) LiH

Figure 2: Energy (Hartree) at different bond length (Å). The solid black line represents the exact energy, while the dotted line corresponds to Hartree-Fock (HF) results. The green and red markers show results from the VQE from circuits sampled from models trained at given bond distances, as indicated in the legend.

To assess the transferability of the circuits with respect to molecular geometry (i.e., varying bond lengths), we employed the circuit composition $\mathcal{G}^*$, which corresponds to the circuits that achieved the lowest energy during the training process for each bond distance and molecule, as shown in Fig. 2. We then re-optimized the parameters of these circuits for bond distances different from those seen during the training phase. This process aims to understand the generalizability of $\mathcal{G}^*$ in accurately representing the molecular system across a broader range of configurations. The results show that the re-optimized circuits consistently maintained chemical accuracy across new bond lengths. This highlights the robustness of the circuit architectures learned by the GFlowNet models, even when applied to bond lengths outside the initial training range.

In addition to accurately solve the ground state energy problem within chemical accuracy, GFlowNets discover quantum circuits that are an order of magnitude smaller than existing approaches. Figure 3

further analyzes the quantum circuit complexity required to estimate molecular ground state energies w.r.t. the number of parameters, gates, and circuit depth. The comparison between circuits sampled by our GFlowNet model, CRLQAS (Patel et al., 2024), QuantumDARTS, and UCCSD for $H_2$ and LiH reveals significant reductions in complexity. Our approach requires significantly fewer parameters, gates, and shallower depths compared to CRLQAS, QuantumDARTS and UCCSD, while maintaining energy estimation within chemical accuracy. The performance gain in terms of gate count and circuit depth for GFlowNet-based sampling highlights the practicality of our method in resource-constrained quantum hardware environments, making it a promising approach for larger molecular systems.

Our method outperforms existing solutions like CRLQAS, QuantumDARTS and UCCSD by achieving comparable energy estimates within chemical accuracy while significantly reducing the quantum circuit complexity. The ability to sample multiple circuits with fewer parameters, gates, and shallower depths makes our GFlowNet-based approach more practical for use in resource-limited quantum hardware environments, offering a scalable solution for larger molecular systems.

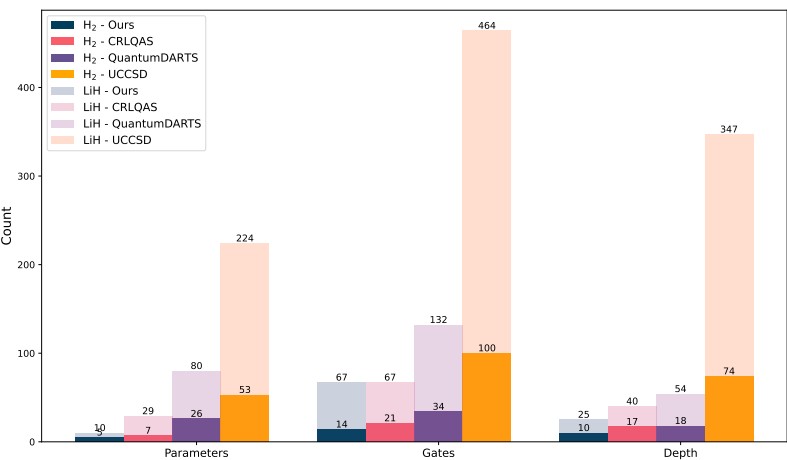

Figure 3: Comparison of computational resource usage of our method, CRLQAS, QuantumDARTS, and UCCSD, across three metrics: number of parameters (equal to the number of rotation gates), number of gates, and circuit depth.

## 6.2 UNWEIGHTED MAX CUT PROBLEM

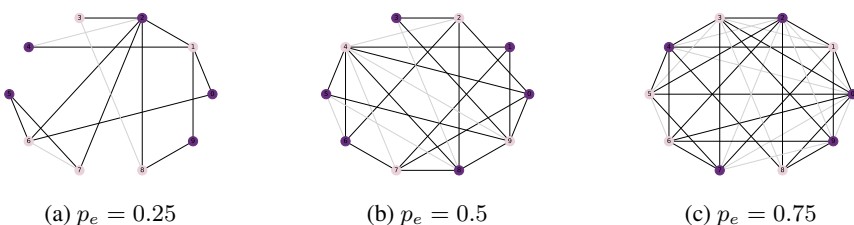

(a) $p_e = 0.25$        (b) $p_e = 0.5$        (c) $p_e = 0.75$

Figure 4: Examples of optimal solutions for the unweighted Max Cut problem. All vertices are divided into two sets, colored light purple and deep purple. The edges between the two sets are shown in black, while the edges within each set are shown in light gray.

We generate random graphs using the Erdős-Rényi model with different probabilities of edge creation $p_e = 0.25, 0.5, 0.75$. The goal is to assess the GFlowNets to search circuits for increasing edge density. As illustrated in Figure 4, the visual representations of these solutions for the probabilities of edge creation $p_e = 0.25$, $p_e = 0.5$, and $p_e = 0.75$, respectively, depict the division of vertices into two sets, with edges between sets shown in black and within sets in light gray. This clear distinction in the graph's structure highlights the effectiveness of GFlowNets in finding optimal cut solutions at different graph densities.

Table 2: Optimal solutions for the unweighted Max Cut problem on graphs generated by the Erdős–Rényi model with 10 vertices. $p_e$ is the probability of edge creation, $N_{edges}$ is the number of edges, D, $D_{min}$, and $D_{max}$ are the mean, minimum, and maximum degree, respectively. CVaR represents the Conditional Value-at-Risk metric, and Depth denotes the depth of the solution quantum circuits. The results are averaged over 10 graphs.

| $p_e$ | $N_{edges}$ | D | $D_{min}$ | $D_{max}$ | CVaR | Depth |
|------|-------------|------------|------------|------------|-------------|---------------|
| 0.25 | 12.80 (2.70) | 2.56 (0.54) | 0.70 (0.67) | 4.40 (0.70) | 11.10 (2.24) | 21.90 (5.36) |
| 0.50 | 22.20 (3.05) | 4.44 (0.61) | 2.10 (0.99) | 6.50 (0.71) | 16.60 (1.90) | 23.70 (10.53) |
| 0.75 | 34.20 (4.10) | 6.84 (0.82) | 4.60 (1.35) | 8.50 (0.70) | 22.50 (1.78) | 25.50 (9.51) |

Table 2 summarizes the quantitative outcomes, showing that as $p_e$ increases, the graph's density and complexity also rise, reflected in the higher number of edges and mean degree. For $p_e = 0.25$, the average number of edges is 12.80, with a mean degree of 2.56, while for $p_e = 0.5$, these values nearly double to 22.20 and 4.44, respectively. At $p_e = 0.75$, the graph exhibits 34.20 edges and a mean degree of 6.84. The CVaR results for our model are 12, 18, and 24 for the respective $p_e$ values in Figure 4, matching the maximal cut values. Our results indicate that GFlowNets not only achieve optimal solutions but also efficiently manage the complexities of quantum circuit design.

We compared our results with those from other works, noting that only QuantumDARTS (Wu et al., 2023) successfully solved the 10-node unweighted Max-Cut problem. However, our method produces more efficient circuits than those reported by QuantumDARTS. On average, our model discovers quantum circuits with depths of 21.90, 23.70, and 25.50 for the respective $p_e$ values, compared to depths of 30, 33, and 34 in their results. This demonstrates that our approach is capable of generating more compact circuits, highlighting its efficiency. Examples of circuits are included in Appendix E.2.

Our method surpasses existing approaches like QuantumDARTS by consistently producing more compact quantum circuits with significantly shallower depths. This demonstrates the efficiency and scalability of our GFlowNet-based approach, making it a more effective solution for tackling the unweighted Max-Cut problem, especially as graph density increases. Our method's ability to handle increasing complexity while maintaining optimal performance highlights its practical advantages in quantum circuit design.

# 7 CONCLUSION

In this work, we demonstrated that GFlowNet-sampled quantum circuits can successfully solve two distinct types of problems: molecular ground state energy estimation and the unweighted Max-Cut problem. For the ground state energy problems of $H_2$ and LiH, our approach consistently yielded energy estimates within chemical accuracy, while requiring significantly fewer parameters, gates, and circuit depth compared to other methods. This reduction in quantum circuit complexity, coupled with the ability to maintain accuracy, makes our method promising for deployment in resource-limited quantum hardware environments, especially for larger molecular systems. Moreover, the generalizability of the circuits generated by GFlowNets, as shown by their successful re-optimization for bond lengths not encountered during training, underscores the robustness of our approach. This capability could be particularly valuable for real-world applications where system configurations may vary. In addressing the unweighted Max-Cut problem, our method produced more efficient circuits with significantly reduced depths compared to QuantumDARTS, while achieving optimal solutions for various graph densities. This highlights the potential of GFlowNets for optimizing quantum circuit designs across combinatorial optimization problems. In future work, we plan to explore the use of pre-trained GFlowNet models that can be fine-tuned for specific tasks, such as quantum error correction and beyond the current scope of Variational Quantum Algorithms. Pre-training models on a broader set of quantum circuits for more quantum problems could potentially enhance generalizability and efficiency. This will allow for faster adaptation to new problem domains. Additionally, we aim to extend the application of GFlowNets to quantum error correction, where designing efficient quantum circuits is crucial for mitigating noise and ensuring the reliability of computations on near-term quantum hardware. Further exploration could also involve applying GFlowNets to other quantum computing tasks.

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

# A  QUANTUM COMPUTATION

In classical computing, a two-state system is represented by a binary state, typically denoted as 0 or 1. The state vector corresponding to such a system can be expressed as $[a, b]^\top$, where $a$ and $b$ represent the probabilities associated with each state. However, in quantum computing, the concept of state is generalized to allow for superposition. A two-state quantum system, or qubit, can exist in a superposition of both classical states (Nielsen & Chuang, 2001). This superposition is described by a state vector in a two-dimensional complex Hilbert space as $|\psi\rangle = \alpha|0\rangle + \beta|1\rangle$, where $|0\rangle$ and $|1\rangle$ are the computational basis states, and $\alpha, \beta \in \mathbb{C}$ are complex coefficients satisfying the normalization condition $|\alpha|^2 + |\beta|^2 = 1$. These coefficients represent amplitudes, and their squared norm correspond to the probabilities of measuring the qubit in the respective basis states.

For a system of $n$ qubits, the dimensionality of the state space increases exponentially, with $d = 2^n$. This means that the system can be described by a superposition of all possible combinations of $n$ qubits.

The evolution of a quantum system can be simulated using quantum circuits. A quantum circuit consists of a sequence of quantum gates, which are unitary operators acting on qubits. Mathematically, the evolution of a quantum state $|\psi\rangle$ under a unitary operator $U$ can be expressed as $|\psi'\rangle = U|\psi\rangle$.

Measuring a quantum system causes a collapse of the superposition state into one of the computational basis states. For example, a single-qubit state $|\psi\rangle = a|0\rangle + b|1\rangle$ will collapse into either $|0\rangle$ or $|1\rangle$ upon measurement. The probability of measuring the state as $|0\rangle$ is given by $|a|^2$, and the probability of measuring $|1\rangle$ is given by $|b|^2$.

## A.1  GATE DECOMPOSITION

On IBM quantum devices, quantum operations are executed using a native gate set $[\mathrm{R_Z}, \mathrm{X}, \mathrm{SX}, \mathrm{ECR}]$ (Chow et al., 2021). These gates form the basis for constructing quantum circuits on this hardware platform. We demonstrate that our selected gate set can be efficiently decomposed into this native gate set, optimizing performance on IBM hardware.

**Hadamard gate**

$$H = \mathrm{R_Z}(\frac{\pi}{2}) \cdot \mathrm{SX} \cdot \mathrm{R_Z}(\frac{\pi}{2})$$

**Rotation-X gate**

$$\mathrm{R_X}(\theta) = \mathrm{R_Z}\left(-\frac{\pi}{2}\right) \cdot \mathrm{SX}^\dagger \cdot \mathrm{R_Z}(\theta) \cdot \mathrm{SX} \cdot \mathrm{R_Z}\left(\frac{\pi}{2}\right)$$

**Rotation-Y gate**

$$\mathrm{R_Y}(\theta) = \mathrm{SX}^\dagger \cdot \mathrm{R_Z}(\theta) \cdot \mathrm{SX}$$

**CNOT gate**

$$\mathrm{CNOT} = (\mathrm{X} \otimes \mathrm{I}) \cdot \mathrm{ECR} \cdot \left(\mathrm{R_Z}\left(-\frac{\pi}{2}\right) \otimes \mathrm{R_Z}\left(-\pi\right) \cdot \mathrm{SX} \cdot \mathrm{R_Z}\left(-\pi\right)\right)$$

## A.2  FORWARD MASK

To enforce constraints on the gate selection process, we define a mask that evaluates the feasibility of applying each quantum gate based on the sequence of previous operations. The mask prevents the model from selecting invalid or redundant actions. Specifically, we define the following rules: (1) A gate cannot be applied consecutively to the same qubit, meaning the selected gate must differ from the previous gate acting on that qubit. (2) Rotation gates from the set $\mathrm{R_X}, \mathrm{R_Y}, \mathrm{R_Z}$ are not allowed to follow another rotation gate, ensuring a diverse set of operations. (3) The gate SX cannot be applied immediately after the gate X, and vice versa, as these gates represent similar operations that could lead to redundancy. (4) A two-qubit gate, such as CNOT, can only be applied if at least one other gate has already been applied to one of the involved qubits, ensuring that multi-qubit operations are meaningful within the context of the quantum state evolution. (5) The model should avoid redundant

two-qubit gates, such as applying a CNOT with control on qubit 0 and target on qubit 1 right after a CNOT with control on qubit 1 and target on qubit 0.

## B    CONNECTION WITH OTHER QAS METHODS

### B.1    DIFFERENTIABLE ARCHITECTURE SEARCH

Gradient-based Quantum Architecture Search algorithms and GFlowNets both rely on the sequential construction of circuits, adding gates step-by-step. However, their exploration and optimization strategies differ significantly. QAS employs a deterministic, task-specific process, iteratively updating circuit parameters and architectures through gradient descent to minimize loss functions, which converge to a single locally optimal solution. In contrast, GFlowNets sample diverse quantum circuit architectures by learning a stochastic policy that balances exploration and exploitation, generating solutions proportional to their rewards. While the sampling process in differentiable architecture search is guided by predefined deterministic functions, GFlowNet's stochastic approach allows it to explore a broader range of effective circuit designs.

### B.2    REINFORCEMENT LEARNING

As discussed in the related work, GFlowNets are formally equivalent to stochastic reinforcement learning (RL) with entropy regularization (Tiapkin et al., 2024). Here, we outline the differences and advantages of GFlowNets compared to traditional RL approaches. Unlike RL, which optimizes a single policy to maximize expected cumulative rewards, GFlowNets are designed to generate diverse, high-reward solutions by directly learning an objective. As noted in (Ostaszewski et al., 2021; Patel et al., 2024), RL methods require the evaluation of intermediate states at each step of a trajectory, and rely heavily on carefully crafted reward functions to guide exploration. In contrast, GFlowNets follow an end-to-end learning approach, bypassing the need for intermediate evaluations and complex reward shaping. This makes them particularly efficient for problems where rewards are defined only at the end of a process, as observables of the problem. These features make GFlowNets particularly adaptive for quantum architecture search problems.

## C    TIME COMPLEXITY ANALYSIS

Here, we discuss the time complexity of the forward process in our method. We define $n$ as the number of qubits, $l$ as the number of circuit layers, $E$ as the embedding dimension, and $L$ as the number of transformer layers in the transformer. As we tokenize quantum circuits, in the worst-case scenario, the length of the sequence is $\mathcal{O}(nl)$. At each step, we use the transformer to compute the logits for each actions, resulting in a complexity of $\mathcal{O}(L(n^2l^2E + nlE^2))$. For a single trajectory, we must compute attention at most $nl$ times. Thus, the worst-case time complexity is $\mathcal{O}(L(n^3l^3E + n^2l^2E^2))$. Considering $T$ iterations, the overall time complexity becomes $\mathcal{O}(TL(n^3l^3E + n^2l^2E^2))$.

For comparison, as reported in (Wu et al., 2023), QuantumDARTS has a time complexity of $\mathcal{O}(4^n lT)$ due to the computational cost associated with the multiplication of unitary matrices.

## D    GROUND STATE ENRGY PROBLEM

Figure 5 illustrates the training rewards over episodes for two models, each with different maximum parameter limits: 10 (red) and 20 (blue). The solid lines represent a moving average over 10 episodes, providing a clearer view of the general trend, while the transparent lines depict the raw, unsmoothed data. Both models show an increase in reward during the early episodes, which stabilizes after approximately 6000 episodes. The model with the higher parameter limit (20) exhibits greater variance due to more tunable parameters.

Figure 6 presents the energy estimates as a function of the number of parameters for models with maximum parameter limits of 5 (green), 10 (brown), and 20 (blue). The plot demonstrates that increasing the number of parameters generally leads to more accurate energy estimates, with most values falling below the chemical accuracy threshold (represented by the black dot-dash line). The models with 10 and 20 parameters show results well within chemical accuracy. Notably, the samples

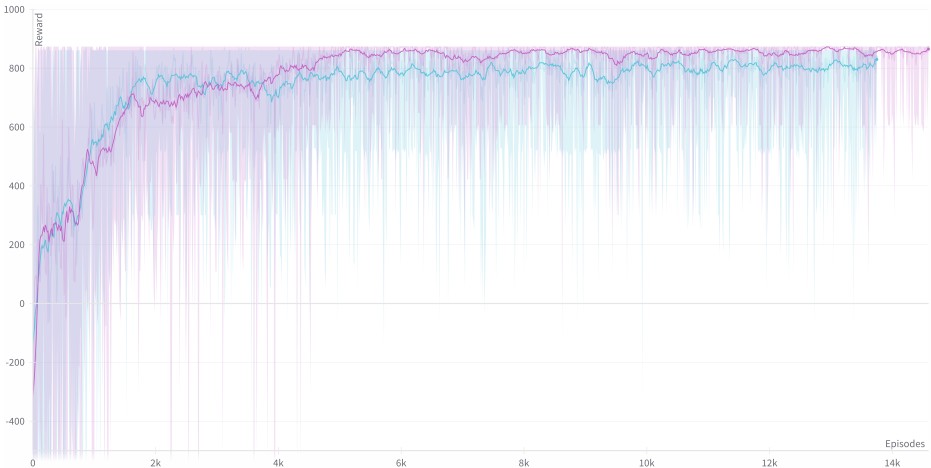

Figure 5: Training rewards over episodes for two models with maximum parameter limits: 10 (red), and 20 (blue) for LiH molecule. The solid lines show a moving average over 10 episodes, while the transparent lines represent the raw, unsmoothed data.

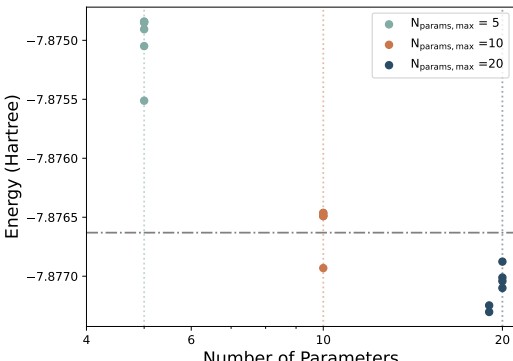

Figure 6: Energy (Hartree) vs. Number of Parameters for different maximum parameter settings for LiH molecule. The plot shows the energy values for three maximum parameter limits: 5 (green), 10 (brown), and 20 (blue). Each group of points, sampled from the models, corresponds to the respective maximum parameter limits, with vertical dashed lines indicating these limits. The black dot-dash line represents the chemical accuracy threshold.

from models with a maximum parameter limit of 20 do not fully utilize all 20 parameters, suggesting that the imposed limits are not always reached, and fewer parameters may still be sufficient to achieve accurate energy estimates.

The figure 7 shows the error values are below chemical accuracy for all bond length, which shows the robustness of the cicuits under varying molecular configurations.

# E    UNWEIGHTED MAX CUT PROBLEM

## E.1    CONDITIONAL VALUE AT RISK

We give the definition of Conditional Value at Risk (CVaR, Barkoutsos et al. (2020)) here. For a random variable $X$ with a confidence level $\beta \in (0, 1]$, and $F_X$ a cumulative distribution function of $X$, the CVaR is defined as,

$$\mathrm{CVaR}_\beta(X) = \mathbb{E}\left[X \mid X \leq F_X^{-1}(\alpha)\right].$$

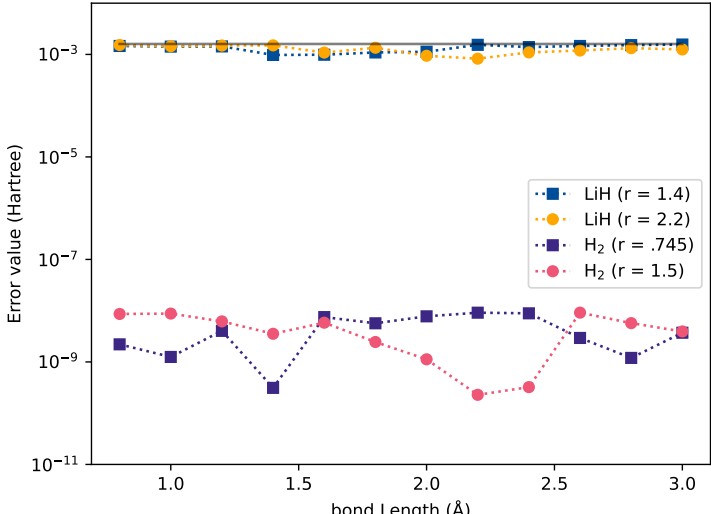

Figure 7: Error values (in Hartree) for LiH and $H_2$ molecules across varying bond lengths.

## E.2 EXAMPLE CIRCUITS

Here, we visualize three example circuits corresponding to solutions of Max-Cut problems.

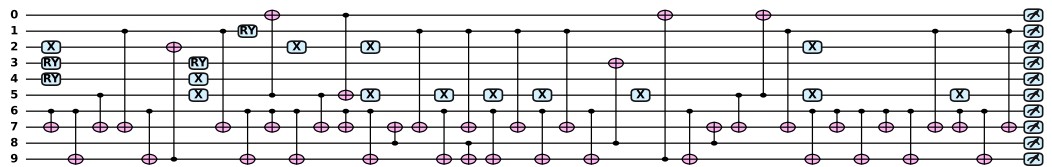

Figure 8: Circuit generated for the example with $p_e = 0.25$.

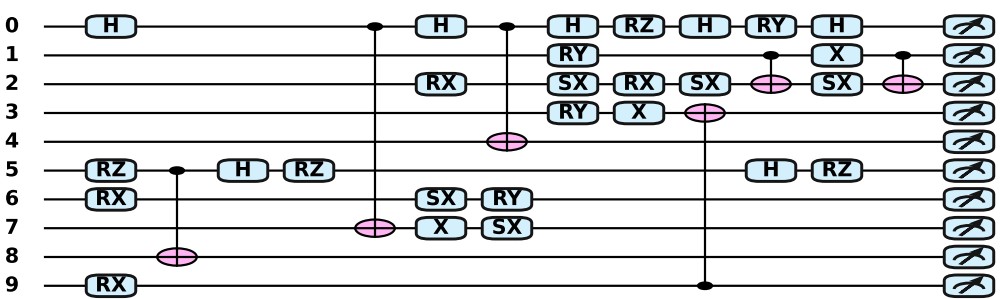

Figure 9: Circuit generated for the example with $p_e = 0.50$.

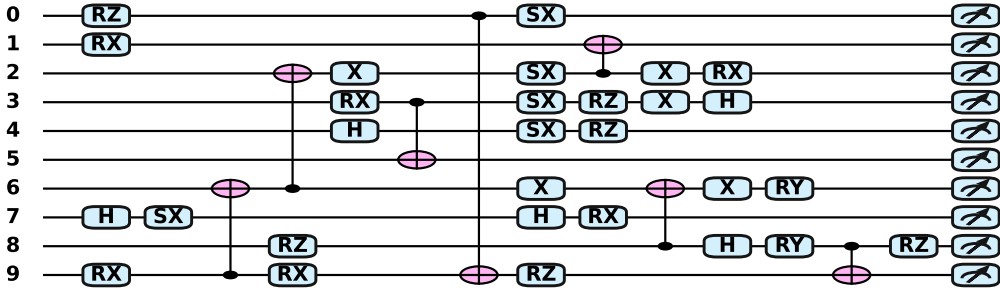

Figure 10: Circuit generated for the example with $p_e = 0.75$.

