# OpenReview forum: "Enhancing Variational Quantum Algorithms: Effective Quantum Ansatz Design Using GFlowNets"
_ICLR.cc/2025/Conference — Submitted to ICLR 2025_

### Official Review · Reviewer_Tkhm · 2024-10-30

**Soundness:** 3
**Presentation:** 2
**Contribution:** 3
**Rating:** 5
**Confidence:** 3

**Summary:**

This paper proposes to use GFlowNets (GFNs) to design the ansatz of variational quantum algorithms (VQAs). The authors apply their quantum architecture search (QAS) framework to ground state energy estimation and the unweighted max-cut problems. The results indicate that for ground state energy estimation, GFNs achieve chemical accuracy while using fewer parameters, gates, and with smaller circuit depths than existing methods such as QuantumDARTS and the UCCSD ansatz. For unweighted max-cut, their approach solves 10-node instances with smaller circuit depths than QuantumDARTS.

**Strengths:**

- To my knowledge, the application of GFlowNets to variational quantum circuits is novel.
- The experimental results show a substantial advantage over existing methods, particularly for the ground state energy problem.
- The generalizability of GFNs is nice to have, although there are no quantitative results presented in comparison to other methods for quantum architecture search.

**Weaknesses:**

- While the experiments show that QAS does better than other models, the problems are relatively small and it is somewhat unclear how well this approach scales for larger problems.
- The use of GFNs doesn't seem to take into account much (if any) problem structure, such as symmetries. Many VQAs leverage permutation symmetry to reduce the size of the search space and achieve better results.
- Overall, this paper could be considered incremental as it simply uses a different ML model for VQAs to obtain improved performance. Among all the other papers along this line, there isn't much in this paper that particularly stands out.
- Several minor grammatical issues and typos throughout the paper.

**Questions:**

- For unweighted max-cut, why was conditional value-at-risk (CVaR) used rather than more common metrics, such as the average and maximum size of the cut found? CVaR seems to be a rather unusual metric, and it would be useful to also provide the operational meaning of this metric.

---

> ### Author Response · Authors · 2024-11-21
>
> I'll address each weakness and questions point by point:
> >While the experiments show that QAS does better than other models, the problems are relatively small and it is somewhat unclear how well this approach scales for larger problems.
>
> While the presented experiments focus on standard benchmarks (e.g., 4 and 6 qubits) and we currently do not have access to a physical quantum computer, these are widely accepted in the community to validate new methods due to their analytical tractability and availability of baseline comparisons. However, the scalability of our approach is supported by the time complexity analysis in Section C, which demonstrates that our method scales polynomially with the number of qubits ($n$) and circuit depth ($l$). This is a significant advantage over methods like QuantumDARTS, which scale exponentially ($\mathcal{O}(4^n l T)$) with the number of qubits. We are actively working on extending our experiments to larger qubit systems, and results on these are planned for future work.
>
> >The use of GFNs doesn't seem to take into account much (if any) problem structure, such as symmetries. Many VQAs leverage permutation symmetry to reduce the size of the search space and achieve better results.
>
> We appreciate this insightful observation. While our current implementation focuses on general-purpose quantum architecture search, we recognize the potential benefits of incorporating problem-specific structures, such as symmetries, to further optimize the search space and improve results. Notably, GFlowNets are well-suited for such enhancements, as they can seamlessly integrate these features, as demonstrated in [1,2].
>
> >Overall, this paper could be considered incremental as it simply uses a different ML model for VQAs to obtain improved performance. Among all the other papers along this line, there isn't much in this paper that particularly stands out.
>
> We respectfully disagree with the characterization of this work as incremental. GFlowNets represent a significant departure from traditional gradient-based or reinforcement learning approaches to QAS. By learning a generative policy that samples solutions proportional to their rewards, GFlowNets address key challenges in quantum architecture search: diversity of solutions, efficient exploration of combinatorial spaces, and end-to-end learning without measuring the intermediate states. Our experiments demonstrate that this approach reduces circuit complexity by an order of magnitude while maintaining or improving accuracy. To further emphasize the novelty of our approach, we will add section B to highlight the limitations of existing methods and clarify the unique advantages of using GFlowNets for QAS.
>
> >Several minor grammatical issues and typos throughout the paper.
>
> Thank you for pointing this out. We will thoroughly proofread the manuscript to address any remaining grammatical issues and typos to ensure clarity and professionalism.
>
> >For unweighted max-cut, why was conditional value-at-risk (CVaR) used rather than more common metrics, such as the average and maximum size of the cut found? CVaR seems to be a rather unusual metric, and it would be useful to also provide the operational meaning of this metric.
>
> The reason we use this metric is to make direct comparisons with other methods, as the lack of publicly available implementations limits comparisons in other metrics.
>
> We are grateful for the reviewer’s thoughtful feedback and constructive questions, which have helped us identify areas where additional clarification and discussion are necessary. If the reviewer finds our responses and proposed updates satisfactory, we kindly ask to reconsider the rating of our manuscript.
>
> [1] Baking Symmetry into GFlowNets, arXiv:2406.05426
>
> [2] Geometric-informed GFlowNets for Structure-Based Drug Design, MoML 2024 Spotlight

---

> ### Author Response · Authors · 2024-11-25
>
> Dear Reviewer Tkhm,
>
> Thank you for your insightful comments on our paper. As the discussion phase deadline approaches, we hope our responses have satisfactorily addressed your concerns. If there are any additional points that require clarification, we would be happy to provide further responses.
>
> Best regards,
>
> The Authors

---

> > ### Comment · Reviewer_Tkhm · 2024-11-25
> >
> > I thank the authors for (partially) addressing my concerns, particularly related to scalability and the relationship to existing works on QAS. Upon taking a look at the revisions made in the supplementary material, I have updated my score to 5.
> >
> > My main concern is still about the scalability; not in terms of how long the algorithm takes to run, but the quality of the solution it finds as the system size increases. Of course I do not expect these circuits to be run at large scale on real devices at the moment, but the 4 and 6 qubit experiments are still very small, and I suppose these numerical experiments can be pushed to 8 or 10 qubits. Also, it would be nice to at least define CVaR within the paper to make it self-contained, rather than forcing the reader to find the definition in the QuantumDarts paper.

---

> > > ### Author Response · Authors · 2024-11-25
> > >
> > > Thank you for your updated score and for acknowledging the revisions we made to address your concerns regarding scalability and the relationship to existing works on quantum architecture search. We appreciate your constructive feedback, which has helped refine the clarity and scope of our work.
> > >
> > > Regarding your main concern about scalability and the quality of solutions as system size increases, we would like to clarify that our numerical experiments for the MaxCut problem were indeed conducted with 10 qubits, as detailed in the main text and supplementary material. These experiments demonstrate the ability of our approach to generate high-quality circuits for this system size, offering evidence that our methods offer better solution than QuantumDarts paper.
> > >
> > > The goal of further extending the experiments is to worked on the larger systems with real devices. However, as noted in the paper, limitations in computational resources and access to physical quantum devices currently constrain our ability to simulate larger problem instances. That said, we believe the demonstrated results at 10 qubits, coupled with our theoretical scalability analysis, provide a solid foundation for confidence in the method’s applicability to larger systems.
> > >
> > > Thank you again for your thoughtful feedback, and we hope these clarifications further address your concerns. If there are any additional questions or suggestions, we would be happy to address them.

---

> > > ### Author Response · Authors · 2024-11-25
> > >
> > > We would also like to highlight that the global optimization of arbitrary quantum circuits is known to be QMA-hard [1, 2, 3]. Despite this complexity, prior work [1] has shown that the use of carefully designed heuristics can significantly reduce gate counts in large-scale quantum circuits, often yielding substantial improvements without requiring a globally optimal solution. This observation serves as a key motivation for our efforts to develop AI-assisted quantum circuit search methods, which aim to achieve similar efficiency gains by leveraging advanced generative models like GFlowNets to explore the solution space effectively. We would like to highlight that numerous studies on GFlowNets have demonstrated their effectiveness in sampling within combinatorial spaces, achieving notable success in various domains such as graph problems [4], language posterior inference [5], and molecular search [6].
> > >
> > > [1] Discretized Quantum Exhaustive Search for Variational Quantum Algorithms, arXiv:2407.17659
> > >
> > > [2] Even Shorter Quantum Circuits for Phase Estimation on Early Fault-Tolerant Quantum Computers with Applications to Ground-State Energy Estimation, PRX Quantum 4, 020331
> > >
> > > [3] Optimising Quantum Circuits is Generally Hard, arXiv:2310.05958
> > >
> > > [4] Let the Flows Tell: Solving Graph Combinatorial Optimization Problems with GFlowNets, NeurIPS 2023 spotlight
> > >
> > > [5] Amortizing intractable inference in large language models, ICLR 2024 oral
> > >
> > > [6] Flow Network based Generative Models for Non-Iterative Diverse Candidate Generation, NeurIPS 2021

---

> > > ### Author Response · Authors · 2024-11-25
> > >
> > > We also included the definition of CVaR in the paper as in Section E.1. Thank you again for your thoughtful feedback, and we hope these clarifications further address your concerns.

---

### Official Review · Reviewer_z4Qj · 2024-11-03

**Soundness:** 2
**Presentation:** 3
**Contribution:** 2
**Rating:** 3
**Confidence:** 4

**Summary:**

In this paper, the quantum circuits are tokenized as input for a sequence model. GFlowNets, along with trajectory balance, are applied to find the optimal composition of a basis gate set to minimize the loss function of a certain task. Such an approach can find a better circuit architecture with fewer gates and shorter depth for variational tasks, including ground state energy and unweighted max cut.

**Strengths:**

1.	Introduce GFlowNets into quantum circuit architecture search as its close relationship with DAGs and state transform;
2.	Utilize a forward mask to filter out redundant gates;
3.	Achieve chemical precision and great transferability for ground state energy problems, and exhibit a major reduction in circuit depth, gate counts, and the number of parameters for max cut problems

**Weaknesses:**

1.	This paper applies the trajectory balance due to faster credit assignment and robustness to long trajectories. However, this paper lacks a convincing explanation of how these two features contribute to the generation of effective quantum circuit ansatz. Also, this paper should provide circuit generated with GFlowNets using other objective functions, such as the original flow matching loss mentioned in the paper, subtrajectory balance objective (Madan, K., Learning GFlowNets From Partial Episodes For Improved Convergence And Stability), and another objective function which deals with extra long trajectories (Pan, L., Better Training of GFlowNets with Local Credit and Incomplete Trajectories).
2.	This paper highlights that the ansatz generation method could result in fewer parameters, gate counts, and depths and deals with scalability problems. However, the experiment circuits are relatively small, and the maximum circuit scale is only 10 nodes in the max cut problem. The authors should consider adding larger problems as related research has demonstrated its circuit search with more than 100 qubits (Wang, H., QuantumNAS: Noise-Adaptive Search for Robust Quantum Circuits).
3.	This paper does not take into consideration hardware constraints, including hardware topology and noise, which are crucial to NISQ devices. These constraints may lead to an increase in circuit depth and gate counts.

**Questions:**

1.	When discussing transferability, Figure 2 only shows the comparison results between VQE-GFNs and HF, how is the transferability of other generated circuits, such as UCCSD, QuantumDARTS, QCAS, and DQAS as shown in Table 1?
2.	In Table 1, all methods except QCAS achieve a result within the chemical accuracy, and “Ours” does not get the smallest error. However, in Figure 3, the “Ours” method achieves a parameter count smaller than the maximum limit. Does that mean the training process should not stop here, or are certain masks too strict?
3.	In Figure 3, 7, 8, and 9, all circuits achieve a parameter count smaller than the parameter limit set by the training process. Will certain experiments return a circuit which has exactly the number of parameters as the limit? As the problem grows more complicated, can the model always return a circuit that is below the parameter number limit?

---

> ### Author Response · Authors · 2024-11-21
>
> I'll address the concerns and questions raised by the reviewer point by point:
>
> >This paper lacks a convincing explanation of how these two features contribute to the generation of effective quantum circuit ansatz. Also, this paper should provide circuit generated with GFlowNets using other objective functions.
>
> While our current work utilizes the trajectory balance objective, we acknowledge the importance of exploring alternative objectives, such as flow matching loss and subtrajectory balance, as mentioned by the reviewer. These objectives offer valuable perspectives, particularly for cases involving long or incomplete trajectories. We plan to conduct a comprehensive comparison of these objectives in future work to evaluate their impact on the generation of effective quantum circuit ansatz.
>
> Our choice to use the TB objective (https://arxiv.org/pdf/2201.1325) is based on its demonstrated empirical advantages, including faster convergence, improved sample diversity, more efficient credit assignment, and robustness to large action spaces—key considerations for our application. We will include this justification in the main text for greater clarity.
>
> While ablating over different GFlowNet objectives would provide additional insights, it is not yet standard practice in the GFlowNet literature. Most papers adopt the TB objective as a default (e.g., arxiv:2310.08774, arxiv:2310.03579), which we have followed here. Additionally, the subtrajectory balance (subTB) objective, while valuable for very long trajectories (as shown in Learning GFlowNets From Partial Episodes For Improved Convergence And Stability), is less relevant for the relatively shallow quantum circuits considered in this work. However, we appreciate the suggestion and will consider these alternative objectives in future iterations of our research.
>
>
> >This paper highlights that the ansatz generation method could result in fewer parameters, gate counts, and depths and deals with scalability problems. However, the experiment circuits are relatively small, and the maximum circuit scale is only 10 nodes in the max cut problem. The authors should consider adding larger problems as related research has demonstrated its circuit search with more than 100 qubits (Wang, H., QuantumNAS: Noise-Adaptive Search for Robust Quantum Circuits).
>
> We note that our experiments focus on small, standard benchmarks commonly used in VQA research for proof of concept and comparative analysis, as they provide a controlled environment for validating new methods. That said, we agree that scalability to larger systems is essential. While our theoretical time complexity analysis demonstrates polynomial scaling, empirical validation on larger systems remains a critical next step. Unfortunately, the lack of access to a physical quantum computer currently limits our ability to perform such validation. We would also like to clarify that the reference cited benchmarks problems with up to 27 qubits, which are already challenging to simulate on classical devices. Furthermore, their method exhibits exponential scaling with the number of qubits, as shown in their time complexity analysis, which fundamentally constrains its applicability to larger problems. In contrast, our method’s polynomial scaling offers a clear path toward addressing larger and more complex quantum systems.
>
> >This paper does not take into consideration hardware constraints, including hardware topology and noise, which are crucial to NISQ devices. These constraints may lead to an increase in circuit depth and gate counts.
>
> We appreciate the importance of hardware constraints in practical quantum computing. While our current framework operates in a noise-free simulation environment, it is designed to be adaptable. By incorporating forward masks or hardware-aware reward functions, we can avoid those incompatible with specific topologies. This adaptability makes the approach suitable for deployment on NISQ devices. In future work, we plan to extend the method to explicitly account for noise and hardware topology, ensuring the generated circuits are not only efficient but also practical for implementation on real devices.

---

> > ### Author Response · Authors · 2024-11-21
> >
> > >When discussing transferability, Figure 2 only shows the comparison results between VQE-GFNs and HF, how is the transferability of other generated circuits, such as UCCSD, QuantumDARTS, QCAS, and DQAS as shown in Table 1?
> >
> > Figure 2 highlights VQE-GFNs to demonstrate the transferability of circuits generated by our approach, specifically focusing on their generalizability to unseen bond lengths during training. UCCSD, as a standard chemistry ansatz, is inherently designed to be transferable. Unfortunately, for other methods, such as QuantumDARTS and QCAS, publicly available implementations are not provided, limiting direct comparisons in this context.
> >
> > >In Table 1, all methods except QCAS achieve a result within the chemical accuracy, and “Ours” does not get the smallest error. However, in Figure 3, the “Ours” method achieves a parameter count smaller than the maximum limit. Does that mean the training process should not stop here, or are certain masks too strict?
> >
> > The fact that the parameter count is below the limit does not necessarily indicate premature stopping. Instead, it reflects the model’s ability to identify efficient circuits without fully utilizing the parameter budget. The masks enforce logical constraints to avoid redundant or invalid gate selections. It is important to note that all benchmarks on VQE focus on finding the most efficient quantum circuits that achieve chemical accuracy, as this level of precision is essential for making realistic and reliable chemical predictions.
> >
> > >In Figure 3, 7, 8, and 9, all circuits achieve a parameter count smaller than the parameter limit set by the training process. Will certain experiments return a circuit which has exactly the number of parameters as the limit? As the problem grows more complicated, can the model always return a circuit that is below the parameter number limit?
> >
> > While the model generally returns circuits below the parameter limit due to its efficiency-focused design, it is possible for the parameter count to approach the limit for more complex problems. This flexibility demonstrates the model’s adaptability, as it tailors circuit complexity to the problem’s requirements. For larger and more complex problems, the model can dynamically utilize the parameter budget while still prioritizing efficiency.
> >
> > We greatly appreciate the reviewer’s detailed insights and thoughtful questions, which have helped us identify key areas for clarification and improvement. If the reviewer is satisfied with our responses and proposed revisions, we kindly request to re-evaluate the rating of our manuscript.

---

> > > ### Comment · Reviewer_z4Qj · 2024-11-24
> > >
> > > The author's feedback on my last two questions addressed those concerns well. However, the paper still needs to improve a lot to be publishable at ICLR. I agree the idea of the paper could have more space to be improved, as I mentioned in my previous comments. However, I have to keep my negative score for the current version.

---

> > ### Comment · Reviewer_z4Qj · 2024-11-24
> >
> > QuantumNAS does not offer experiments over 27 qubits in their paper as it is a 2020 paper, but the related repo Torchquantum, it already enables 100 qubits search capability. More recent research, 'Élivágar: Efficient quantum circuit search for classification,' also demonstrates polynomial scaling, and they do consider the hardware limitations and constraints which seems more advanced compared to the results in this paper.

---

> > > ### Author Response · Authors · 2024-11-24
> > >
> > > Could you provide any sources to support the claim that “it already enables 100 qubits search capability”? We could not find evidence of this in the repository or anywhere. QuantumNAS, their most recent work on this topic as of 2022, includes benchmark problems with up to 21 qubits (not 27 qubits—apologies for the earlier typo).
> > >
> > >
> > > [1] QuantumNAS: Noise-Adaptive Search for Robust Quantum Circuits, HPCA-28.

---

> > > > ### Comment · Reviewer_z4Qj · 2024-11-25
> > > >
> > > > Apologize for my misleading, the scale is not meant for qubits but for the number of parameters - both of your technologies can be implemented on real quantum hardware no matter how many qubits you have - simulation results in terms of the number of qubits depend on your hardware memory side.

---

> ### Author Response · Authors · 2024-11-25
>
> Thank you for acknowledging that our responses addressed your concerns. However, we would like to clarify and address the points raised:
>
> > More recent research, 'Élivágar: Efficient quantum circuit search for classification,' also demonstrates polynomial scaling, and they do consider the hardware limitations and constraints which seems more advanced compared to the results in this paper.
>
> We respectfully disagree with the reviewer’s comparison. For the work “Élivágar: Efficient Quantum Circuit Search for Classification”, they focus on quantum machine learning (QML), they use the methods based on random sampling with two metric to post selecting circuits, Clifford noise resilience, and Representational capacity which is tailored to QML problems. The work focus on the QML problems, and they are not designed for the broader class of general quantum computing problems that our work addresses.
>
> Moreover, rather than contradicting our approach, Élivágar supports the feasibility of incorporating hardware-aware features, as these can seamlessly be integrated into the sampling process or reward functions within the GFlowNet framework. This adaptability is a key strength of GFlowNets.
>
> We also note that Élivágar includes comparisons with QuantumNAS. However, we would like to emphasize again that QuantumNAS is not considered a state-of-the-art method in this domain. More recent work, such as QuantumDARTS (ICML 2023) and CRLQAS (ICLR 2024), as highlighted by another reviewer, represents the current state of the art. Both of these works utilize benchmarks similar to ours, making them more relevant points of comparison. Notably, our method demonstrates superior performance in designing efficient circuits compared to these state-of-the-art approaches.
>
> >Apologize for my misleading, the scale is not meant for qubits but for the number of parameters
>
> We agree that the reviewer’s initial comment misrepresented the scalability of the current research in this field. The clarification that the “scale” referred to the number of parameters rather than qubits highlights a misunderstanding that may have led to an inaccurate assessment of our work’s scope and contributions.
>
> We appreciate the clarification but believe this initial misrepresentation should be taken into account when evaluating the validity and fairness of the comments.
>
> > - both of your technologies can be implemented on real quantum hardware no matter how many qubits you have - simulation results in terms of the number of qubits depend on your hardware memory side.
>
> We would like to emphasize again that QuantumNAS relies on the unitary of the circuits to evaluate metrics for co-searching. This requirement inherently leads to exponential scaling with the number of qubits, which significantly limits its applicability to large systems. This limitation contrasts with our approach, which demonstrates polynomial scaling, making it more suitable for addressing larger and more complex quantum systems.
>
> Furthermore, this limitation of QuantumNAS is also acknowledged and supported by the work Élivágar, as cited by the reviewer, which highlights the challenges of scaling such methods to larger qubit systems.
>
> >However, the paper still needs to improve a lot to be publishable at ICLR. I agree the idea of the paper could have more space to be improved, as I mentioned in my previous comments. However, I have to keep my negative score for the current version.
>
> We are disappointed that despite recognizing the addressed concerns and the potential of the ideas presented in the paper, you have chosen to maintain a negative score without providing additional constructive feedback to justify this decision. If you believe there are further areas that require improvement to meet the standards of ICLR, we would appreciate specific feedback outlining these issues. A blanket statement that “the paper still needs to improve a lot” without elaboration does not provide actionable guidance, particularly when our responses have already addressed the points you raised.
>
> We respectfully request a reconsideration of your evaluation, as we believe the paper presents a strong and original contribution that aligns with the expectations of the conference. Thank you for engaging in this discussion and for your time reviewing our work.

---

> > ### Comment · Reviewer_z4Qj · 2024-11-25
> >
> > I believe the scalability issue is not only raised by myside, and it also does not mean the QuantumNAS should still be considered as SOTA work since that work will be completed in 2021. I am not just giving general comments to encourage authors to improve the paper - I do give several suggestions in my previous comments, a good paper does not only provide a reasonable idea, but you need to have a good evaluation, too. For your evaluation part, VQE only supported by H2 and LiH, which are tiny molecules, and for the QAOA part, unweighted 10 nodes graph is also too small to convincing reviewers. For the paper you mentioned, e.g. QuantumNAS and CRLQAS, they do have much more evidence in their evaluation session, at least both of them consider the noisy environment - noisy simulation is much more expansive compared to the noiseless, QuantumNAS involves the real machine experiments, and CRLQAS did a scaling weight to convince the reviewer's concern for the drift may occurs in the quantum environment. Authors cannot just using "adaptive ability" of the proposed model to convince the reviewer, even we can imagine such model could performs good on for example a noise model, authors need to provide the evaluation. Furthermore, all of the references paper QuantumNAS, CRLQAS, QuantumDARTS using the IBM backend, and now, the IBM backend is shifted from fixed frequency qubits to the tunable coupler on fixed qubits architecture, which means the control mechanism is changed and some of the noise resources also changes, if authors could take those important factors from actual quantum side to the evaluation to demonstrate the scalability and adaptive ability, I am more than happy to give a stronger support for the paper.

---

> > > ### Author Response · Authors · 2024-11-26
> > >
> > > We would like to kindly ask the reviewer to provide any sources to support the claim "all of the references paper QuantumNAS, CRLQAS, QuantumDARTS using the IBM backend".

---

> ### Comment · Reviewer_z4Qj · 2024-11-27
>
> CRLQAS considers part of the hardware constraints of IBM_Mumbai in their paper, and they use a scaling weight on the noise model of IBM_Mumbai to provide evidence of their method could "adapt" the noise drift. QuantumDARTS is NOT using a noise model from the IBM backend but a readout error model with data obtained from Sycamore and Zuchongzhi. QuantumNAS consider multiple backends from IBM. Those are all you can easily find in their paper.
>
> Based on your model and your rebuttal, I don't think it's too hard for you to demonstrate your model's results with the consideration of at least a noise model from any of the hardware inventor's backends. At least a runtime varies by the number of qubits with the same noise model could help reviewers re-evaluate the scalability of your backend.
>
> My major point is all of those papers at least include parts of the hardware constraints into consideration, the mechanism shift on the IBM side is one of the interesting topics I recommend to you and is the topic what I think your model could demonstrate your model's advantage on.

---

> > ### Author Response · Authors · 2024-11-27
> >
> > We appreciate your continued engagement and detailed feedback. Below, we address your comments based on the mutually agreed-upon facts and the constraints of our work.
> >
> > 1. Backend Usage in Referenced Works
> >
> > We mutually agree that ONLY QuantumNAS conducted experiments on IBM backends. The other referenced works, CRLQAS and QuantumDARTS, did not utilize IBM hardware; CRLQAS incorporated noise models based on the data of IBM_Mumbai for systems up to 4 qubits but performed simulations using the Pauli transfer matrix (PTM). QuantumDARTS conducted noise-free experiments up to 10 qubits, similar to our MaxCut benchmarks, and performed noisy experiments exclusively on QML tasks using readout error models.
> >
> > 2. Experimental Scale and Comparability
> >
> > While we mutually recognize the contributions of CRLQAS and QuantumDARTS, their experimental scales align closely with ours. CRLQAS performed noisy simulations up to 4 qubits and noise-free simulations up to 8 qubits. QuantumDARTS executed noise-free experiments up to 10 qubits, similar to our MaxCut benchmarks, and conducted noisy experiments only on QML-specific tasks up to 8 qubits. Our work’s experimental scale, including noiseless evaluations of 10-node MaxCut and noise-free VQE tasks (e.g., H$_2$ and LiH), is directly comparable to both.
> >
> > We agree that scaling to larger systems and incorporating noise models from quantum devices are crucial for demonstrating broader applicability. However, these extensions must be balanced against the computational challenges of simulating noisy quantum systems. It is well known that simulating noisy qubits is computationally resource-intensive, scales poorly with system size, and involves trade-offs between the accuracy of the noise model and the types of operations that can be simulated [1].
> >
> > 3. Practical Constraints on Real-Device Access
> >
> > We also appreciate your suggestion to explore runtime variability under fixed noise models as a means to demonstrate scalability. This is a valuable direction, and we will incorporate it into our ongoing efforts. Currently, we lack access to real quantum hardware, which limits our ability to perform experiments on physical devices. Nevertheless, our framework is designed to accommodate hardware-specific constraints, such as topology and noise, by modifying the sampling process or reward function. We plan to explore these directions in future work, leveraging noise-aware simulators or real-device access as resources become available.
> >
> >
> > [1] Noisy gates for simulating quantum computers, Phys. Rev. Research 5, 043210 (2023).

---

### Official Review · Reviewer_7NCk · 2024-11-03

**Soundness:** 2
**Presentation:** 2
**Contribution:** 2
**Rating:** 3
**Confidence:** 5

**Summary:**

This paper deployed a generative model (GflowNet) to assist the design of variational quantum algorithm. The novelty is relatively weak as it applied a popular techniques of generative machine learning to a quantum application. Quantum architecture search problem has been widely studied by many literature, where the biggest challenge lie on the architecture search space and computational cost. However, the authors do not explore or discuss it deep enough. Meanwhile, the numerical results are relatively weak.

**Strengths:**

Leveraging generative models for quantum algorithms is an interesting topic

**Weaknesses:**

The computational cost is not discussed.
The numerical experiments are weak and are not well reported.
Please see the detailed comments in Questions.

**Questions:**

Here are a few major questions and comments regarding the methodology and numerical experiments:
1. How is the state space designed? How to deal with exponential complexity in the design state? How is the scalability of the approach?
2. How to optimize the ansatz parameters and ansatz architecture simultaneously, especially from the computational cost perspective. Some computational cost should be reported.
3. The sizes of the example cases are very small, e.g., 4 and 6 qubits Hamitonians for ground state energy problem.
4. How many iterations/circuit trails does the framework take to converge?
5. In Figure 3 and Tables 1 & 3, the standard deviation should be reported.
6. In Figure 3, the authors only compare to one architecture search algorithm (QDARTS) whose idea was first proposed in arXiv 2010.08561 and a modified version (Wu et al 2023) is published in ICML 2023. The baseline is not strong enough given that there are many quantum architecture search literatures now.
7. Although the proposed generative model finds better ansatz than the QDARTS, I image QDARTS would be more efficient.

---

> ### Author Response · Authors · 2024-11-21
>
> I'll address the concerns and questions raised by the reviewer point by point:
>
> 1. Regarding the computational cost and state space design:
>
> The state space is designed as a directed acyclic graph where states represent partially constructed quantum circuits, and actions correspond to the selection of quantum gates as shown in Section 4.1. To handle the exponential complexity, we leverage GFlowNets, which effectively sample from a learned distribution proportional to the reward function. This allows efficient exploration of the combinatorial space without requiring exhaustive enumeration. Scalability is ensured through the use of transformers to the stochastic policy, which scale efficiently with the number of qubits and layers, as shown in our time complexity analysis.
>
> 2. On optimizing ansatz parameters and architecture simultaneously:
>
> We appreciate this important point. The ansatz parameters and architecture are optimized sequentially ,not simultaneously, within the GFlowNet framework. During the forward pass, the architecture is sampled, and its parameters are optimized using a classical optimizer to compute the reward as shown in Section 4.1. Our reported time complexity $\left(\mathcal{O}\left(T L\left(n^3 l^3 E+n^2 l^2 E^2\right)\right)\right.$ ) demonstrates the efficiency of this approach compared to baseline methods like QuantumDARTS $\left(\mathcal{O}\left(4^n l T\right)\right.$ ). The computational costs are detailed in the manuscript (Section C).
>
> 3. Concerning the small size of example cases:
>
> We understand the reviewer's concern about the limited size of our example cases. While the examples involve 4 and 6 qubit Hamiltonians, these are standard benchmarks in variational quantum algorithm research. These problems serve as proof of concept and validate the efficiency of GFlowNet-sampled circuits. Future work will focus on scaling the approach to larger systems and demonstrating its applicability to higher qubit counts.
>
> 4. About the number of iterations/circuit trials for convergence:
>
> We apologize for overlooking this important detail. The framework typically converges within approximately 6000 iterations for the problems considered, as illustrated in the training rewards plot (Figure 5). This demonstrates the efficiency of GFlowNets in achieving convergence compared to traditional search methods.
>
> 5. Regarding the standard deviation in results:
>
> We agree that including standard deviations would enhance the reliability of our results. In the revised version, we updated Tables 1 & 3 to include standard deviations for all reported metrics of our methods due to the lack of publicly available implementations of other methods.
>
> 6. On the limited comparison to other architecture search algorithms:
>
> We appreciate this feedback. QuantumDARTS from ICML 2023 was chosen as it represented a state-of-the-art method at the time of our experiments. Based on feedback from other reviewers, we have added comparisons to an additional state-of-the-art method based on reinforcement learning. We would be glad to compare with more methods if the reviewer could provide specific suggestions.
>
> 7. Although the proposed generative model finds better ansatz than the QDARTS, I image QDARTS would be more efficient.
>
> We respectfully disagree that QDARTS would be more efficient. Our method offers better scalability and efficiency compared to QDARTS, especially for larger problem sizes. The GFlowNet approach allows for more efficient exploration of the architecture space, resulting in faster convergence and better-quality solutions. Additionally, our method's ability to generate diverse circuit architectures provides more flexibility in adapting to different problem requirements and hardware constraints.
>
> We appreciate the reviewer’s detailed feedback and constructive suggestions, which will undoubtedly strengthen our work. If the reviewer finds these updates satisfactory, we kindly request to consider revisiting the rating of our manuscript.

---

> ### Author Response · Authors · 2024-11-25
>
> Dear Reviewer 7NCk,
>
> Thank you for your insightful comments on our paper. As the discussion phase deadline approaches, we hope our responses have satisfactorily addressed your concerns. If there are any additional points that require clarification, we would be happy to provide further responses.
>
> Best regards,
>
> The Authors

---

> > ### Comment · Reviewer_7NCk · 2024-11-27
> >
> > Thanks the authors for the efforts and response. I have also read the comments communicated with reviewer z4Qj.
> >
> > The benchmarking and baseline comparison are still the main concern. Moreover, applying another developed ML model to a quantum architecture search application does not sound attractive enough, at least to me. So I intend to keep the original score.

---

> ### Comment · Area_Chair_KLhy · 2024-11-27
>
> Dear Reviewer,
>
> The authors have provided their rebuttal to your comments/questions. Given that we are not far from the end of author-reviewer discussions, it will be very helpful if you can take a look at their rebuttal and provide any further comments. Even if you do not have further comments, please also confirm that you have read the rebuttal. Thanks!
>
> Best wishes,
> AC

---

### Official Review · Reviewer_B4xX · 2024-11-04

**Soundness:** 3
**Presentation:** 3
**Contribution:** 3
**Rating:** 6
**Confidence:** 5

**Summary:**

The authors proposed a novel quantum architecture search algorithm utilizing GFlowNets. The GFlowNet based algorithm is able to generate efficient ansatz tailored for tasks such as the ground state energy estimation and the unweighted max-cut problem. Numerical experiments show that the proposed algorithm is able to reduce the gate count while achieving good accuracy compared to previous literature.

**Strengths:**

1. The essence of GFlowNets actually fits the nature of quantum architecture search, which can be further explored as one of the AI for quantum algorithms.
2. Well-structured article with properly selected experimental tasks to demonstrate the efficiency of the proposed method.

**Weaknesses:**

1. The scalability of the QAS approaches is always a main concern in ensuring the practical utility of these methods. The authors could provide a more detailed discussion about this topic.
2. The results of both bond lengths in Figure 2 are very close to the ground state, which is hard to tell from the figure. It would be better if you could provide the energy error along with the energy value (as in [1]).
3. As QAS is evolving very fast, QuantumDARTS is actually not the SOTA at this time. [2] provided a curriculum reinforcement learning method that surpasses QuantumDARTS on the ground state energy estimation problem. The authors could consider their results.
4. The proposed method is mainly examined by numerical results and lacks theoretical guarantees (with only theorem 4.1).


[1] Experimental quantum computational chemistry with optimized unitary coupled cluster ansatz

[2] Curriculum reinforcement learning for quantum architecture search under hardware errors

**Questions:**

No Questions.

---

> ### Author Response · Authors · 2024-11-21
>
> I'll address each weakness point by point:
>
> 1. Regarding the scalability of our approaches:
>
> Thank you for constructive suggestions. We acknowledge the importance of scalability in ensuring the practical utility of our method. For the QAS problem, the number of search possible gate arrangements grows exponentially with number of qubits and number of layers, which makes the search space large and combinatorial in nature. The motivation of our work is to address this challenge by developing a stochastic search algorithm. Our approach uses GFlowNets to learn stochastic policies to sample the vast search space, allowing us to find optimal quantum circuit architectures. Many works of GFlowNets demonstrate the numerical success of sampling in combinatorial space, including graph problem [1], language posterior inference [2], Molecule search [3]. We add a section to discuss about the complexity of our methods, which shows $\mathcal{O}\left(T L\left(n^3 l^3 E+n^2 l^2 E^2\right)\right)$, comparing to QuantumDARTS time complexity, $\mathcal{O}\left(4^n l T\right)$.
>
> 2. Concerning the results of bond lengths in Figure 2:
>
> We appreciate this suggestion. In the revised manuscript, we will include the energy error along with the energy value (Figure 9 in the paper), similar to the approach in [4]. This addition will provide a clearer representation of our results' accuracy compared to the ground state.
>
> 3. About the current state-of-the-art in QAS:
>
> Thank you for bringing this to our attention. We updated our comparison to include the curriculum reinforcement learning method mentioned in [5]. We will discuss how our approach compares to this more recent method in terms of performance on ground state energy estimation.
>
> 4. On the lack of theoretical guarantees:
>
> We acknowledge the importance of establishing strong theoretical foundations. Notably, there is a growing theoretical research on GFlowNets, showing the connections to MCMC [6], reinforcement learning [7], and variational inference [8]. While this work primarily emphasizes numerical results, we plan to explore and expand upon the theoretical aspects in future studies.
>
> We deeply value the reviewer’s thoughtful feedback and insightful questions, which have been instrumental in identifying areas for improvement and further clarification. If our responses and suggested revisions address the concerns effectively, we kindly invite the reviewer to reconsider their rating of our manuscript.
>
> [1] Let the Flows Tell: Solving Graph Combinatorial Optimization Problems with GFlowNets
>
> [2]  Amortizing intractable inference in large language models
>
> [3]Flow Network based Generative Models for Non-Iterative Diverse Candidate Generation
>
> [4] Experimental quantum computational chemistry with optimized unitary coupled cluster ansatz
>
> [5] Curriculum reinforcement learning for quantum architecture search under hardware error
>
> [6] Generative Flow Networks: a Markov Chain Perspective
>
> [7] Generative Flow Networks as Entropy-Regularized RL
>
> [8] GFlowNets and variational inference

---

> > ### Comment · Reviewer_B4xX · 2024-11-27
> >
> > Thank you to the authors for their response. I have no further questions, and I intend to keep my score.

---

> ### Author Response · Authors · 2024-11-25
>
> Dear Reviewer B4xX,
>
> Thank you for your insightful comments on our paper. As the discussion phase deadline approaches, we hope our responses have satisfactorily addressed your concerns. If there are any additional points that require clarification, we would be happy to provide further responses.
>
> Best regards,
>
> The Authors

---

> ### Comment · Area_Chair_KLhy · 2024-11-27
>
> Dear Reviewer,
>
> The authors have provided their rebuttal to your comments/questions. Given that we are not far from the end of author-reviewer discussions, it will be very helpful if you can take a look at their rebuttal and provide any further comments. Even if you do not have further comments, please also confirm that you have read the rebuttal. Thanks!
>
> Best wishes,
> AC

---

### Author Response · Authors · 2024-11-21
**General Rebuttal**

We sincerely thank the reviewers for their thoughtful feedback, constructive suggestions, and detailed questions. Their insights have helped us identify key areas for clarification and improvement, which we have addressed in our responses and proposed revisions. We show our changes highlighted in blue color in Supplementary Material.

**Scalability**

Scalability is a critical concern in quantum architecture search, and we have addressed this in multiple ways. The scalability of our approach is demonstrated through theoretical time complexity analysis, as shown in Section C, revealing polynomial scaling with the number of qubits and circuit depth. This represents a significant improvement over methods like QuantumDARTS that scale exponentially. While our current experiments are limited to small benchmarks commonly used in variational quantum algorithm research, they serve as proof of concept. We are actively working to extend our approach to larger systems and validate its empirical scalability in future work.

**Numerical Experiments**

We have addressed concerns about numerical experiments and reporting by including more comprehensive metrics, such as energy errors alongside energy values, and standard deviations for reported metrics of our methods. Comparisons have been updated to include more recent state-of-the-art methods as suggested by reviewers where possible, such as curriculum reinforcement learning approaches.

**Use of GFlowNets**

The use of GFlowNets in our work represents a significant departure from traditional gradient-based or reinforcement learning approaches, as we discuss in Section B. By leveraging their ability to sample solutions proportional to their rewards, GFlowNets enable efficient exploration of combinatorial spaces, generate diverse quantum circuit architectures, and maintain strong end-to-end learning capabilities. While our current implementation does not explicitly incorporate problem-specific structures, such as symmetries, GFlowNets are well-suited for these enhancements, as demonstrated in recent works. This is an exciting direction for future exploration.

---

### Meta-Review · Area_Chair_KLhy · 2024-12-07

**Metareview:**

This paper proposed to improve variational quantum algorithms by GFlowNets. In the numerical experiments, it is demonstrated that GFlowNets can discover ansatz with fewer parameters, gate counts, and depths compared to current approaches for the molecular electronic ground state energy problem.

During the rebuttal, there were adequate discussions between the author and the reviewers. Some perspectives, such as the novelty of GFlowNet and details for numerical experiments were explained. However, the scalability is a general concern with only a few qubits applied in the current experiments. On the one hand, variational quantum algorithms are seen as a counterpart of classical neural networks, which is of much larger size and has been very practical. On the other hand, this work is purely experimental without theory. In all, the result is not convincing enough from the general perspective of machine learning, which is also reflected in the scores. The decision is hence rejection for ICLR 2025.

**Additional Comments On Reviewer Discussion:**

There were adequate discussions during the rebuttal period.

---

### Decision · Program_Chairs · 2025-01-22

Reject